# CrowdSpeech and VoxDIY: Benchmark Datasets for Crowdsourced Audio Transcription

**Nikita Pavlichenko**
Yandex
Moscow, Russia
pavlichenko@yandex-team.ru

**Ivan Stelmakh**
Carnegie Mellon University
Pittsburgh, PA, USA
stiv@cs.cmu.edu

**Dmitry Ustalov**
Yandex
Saint Petersburg, Russia
dustalov@yandex-team.ru

## Abstract

Domain-specific data is the crux of the successful transfer of machine learning systems from benchmarks to real life. In simple problems such as image classification, crowdsourcing has become one of the standard tools for cheap and time-efficient data collection: thanks in large part to advances in research on aggregation methods. However, the applicability of crowdsourcing to more complex tasks (e.g., speech recognition) remains limited due to the lack of principled aggregation methods for these modalities. The main obstacle towards designing aggregation methods for more advanced applications is the absence of training data, and in this work, *we focus on bridging this gap in speech recognition*. For this, we collect and release CROWDSPEECH — the first publicly available large-scale dataset of crowdsourced audio transcriptions. Evaluation of existing and novel aggregation methods on our data shows room for improvement, suggesting that our work may entail the design of better algorithms. At a higher level, *we also contribute to the more general challenge of developing the methodology for reliable data collection via crowdsourcing*. In that, we design a principled pipeline for constructing datasets of crowdsourced audio transcriptions in any novel domain. We show its applicability on an under-resourced language by constructing VOXDIY — a counterpart of CROWDSPEECH for the Russian language. We also release the code that allows a full replication of our data collection pipeline and share various insights on best practices of data collection via crowdsourcing.[1]

## 1 Introduction

Speech recognition is an important research problem that has found its applications in various areas from voice assistants such as Siri or Alexa [20] to call centers [35] and accessibility tools [4]. The research community has been actively developing tools for automated speech recognition [1, 19, 24, 25, 29, 38, 40, 44, 51, 52, and many other works]. As a result, the state-of-the-art methods achieve near-perfect performance [54] on LIBRISPEECH [37] — a famous benchmark to compare speech recognition systems.

---

[1] The code and data are released at https://github.com/Toloka/CrowdSpeech and https://doi.org/10.5281/zenodo.5574585. Our code is available under the Apache license 2.0, and datasets are available under the CC BY 4.0 license.

35th Conference on Neural Information Processing Systems (NeurIPS 2021) Track on Datasets and Benchmarks.

While the technical performance on curated benchmarks is almost perfect, it does not necessarily result in reliable practical performance [45]. Indeed, in real applications, people may use some specific vocabulary or dialects underrepresented in the conventional training data. Thus, blind application of methods trained on the standard benchmarks may result in low accuracy or, perhaps more concerning, discrimination of some subpopulations. For example, a recent study of YouTube's Automatic Captions reveals a difference in accuracy across gender and dialect of the speaker [46].

One approach towards improving the practical performance of speech-recognition systems is to fine-tune the models in these systems on domain-specific ground truth data. Fine-tuning is very important and efficient for the speech-recognition task [8, 53], but the main problem with this approach is the lack of data. Indeed, even datasets that are considered to be very small in the area (e.g., CHiME-6 [49]) contain hours of annotated audios. While getting such an amount of unlabeled data in a speech-focused application may be feasible, annotating this data with the help of expert annotators may be prohibitively expensive or slow.

Recently, crowdsourcing has become an appealing alternative to the conventional way of labeling data by a small number of experts. Platforms like Mechanical Turk (`https://www.mturk.com/`) and Toloka (`https://toloka.ai/`) significantly reduce the time and cost of data labeling by providing on-demand access to a large crowd of workers. Of course, this flexibility comes at some expense, and the main challenge with the crowdsourcing paradigm is that individual workers are noisy and may produce low-quality results. A long line of work [12, 21, 42, 43, 50, 55, and others] has designed various methods to estimate true answers from noisy workers' responses to address this issue in multiclass classification. As a result, crowdsourcing has become an industry standard for image labeling with small and large technology companies using it to improve their services [13].

In speech recognition, however, the annotations obtained from crowd workers are sentences and not discrete labels, which makes the aforementioned classification methods impractical. Unfortunately, the problem of learning from noisy textual responses and other non-conventional modalities is much less studied in Machine Learning and Computer Science communities. One of the obstacles towards solving this problem in a principled manner is the lack of training data: in contrast to the classification setup, worker answers in the speech recognition tasks are high-dimensional, and researchers need a large amount of data to build and evaluate new methods. Therefore, *we focus our work on bridging this gap by constructing and analyzing a large-scale dataset of crowdsourced audio transcriptions*.

At a higher level, *this work also considers the more general challenge of developing the methodology for reliable data collection via crowdsourcing*. In many areas, data is the key resource for research and development. When the costs of getting data are high, it becomes available to only a privileged population of researchers and practitioners, contributing to the overall inequity in the community. Crowdsourcing offers an appealing opportunity to make data collection affordable. However, to take the full benefits of crowdsourcing, the research community needs to develop procedures and practices to take reliable control over the quality of collected data. In this work, we build on our long experience of industrial data collection at Yandex [14, 15, 16] and share resources as well as insights that may benefit researchers and engineers who want to collect reliable data on crowdsourcing platforms.

**Our Contributions** Overall, in this work we make several contributions:

First, we collect and release CROWDSPEECH — the first publicly available large-scale dataset of crowdsourced audio annotations. In that, we obtain annotations for more than 60 hours of English speech from 3,994 crowd workers.

Second, we propose a fully automated pipeline to construct semi-synthetic datasets of crowdsourced audio annotations in under-resourced domains. Using this procedure, we construct VOXDIY — a counterpart of CROWDSPEECH for Russian language.

Third, we evaluate the performance of several existing and novel methods for aggregation of noisy transcriptions on collected datasets. Our comparisons indicate room for improvement, suggesting that our data may entail progress in designing better algorithms for crowdsourcing speech annotation.

Fourth and finally, we release the code to *fully replicate* the data preparation and data collection processes we execute in this work. Additionally, we share various actionable insights that researchers and practitioners can use to fulfill their data collection needs.

The remainder of this paper is organized as follows. We begin with a survey of related work in Section 2. We then construct a pool of speech recordings for annotation in Section 3 and describe

the annotation pipeline in Section 4. We provide an exploratory analysis of our datasets in Section 5 and evaluate existing and novel methods in Section 6. A short discussion of the results is provided in Section 7. Finally, we note that while the discussion in the present paper is centered around speech recognition, this work extends to other applications where textual sequences may be crowdsourced (e.g., optical character recognition [10, 11, 48]).

## 2  Related Work

Several past works investigated the use of crowdsourcing platforms to obtain training data for speech recognition systems. We now discuss the most relevant studies.

**Aggregation Methods** Despite the problem of aggregation of textual responses has been receiving much less attention than aggregation in the context of classification, there are several works in this direction. The first study dates back to 1997 when Fiscus [18] proposed a method called ROVER to combine outputs of multiple speech-recognition systems. Several subsequent works [3, 17, 26, 31] demonstrated the usefulness of this method in the crowdsourcing setup. More recently, two novel methods, RASA and HRRASA, were proposed to aggregate multiple translations of a sentence from one language to another [27, 28]. Despite the fact that these methods were designed in the context of machine translation, they are positioned as general methods for text aggregation and hence apply to our problem. Overall, the three methods mentioned above constitute a pool of available baselines for our problem. In Section 6, we return to these methods for additional discussion and evaluation.

**Data Collection** Novotney and Callison-Burch [36] annotated about thirty hours of audio recordings on MTurk to evaluate the quality of crowdsourced transcriptions by measuring the accuracy of a speech recognition model trained on this data. They concluded that quantity outweighs quality, that is, a larger number of recordings annotated once each led to a higher accuracy than a smaller number of recordings annotated multiple times with subsequent aggregation. We note, however, that this conclusion may itself be influenced by an absence of a principled aggregation algorithm. Additionally, it is not always the case that one can obtain more data for annotation. Thus, in our work, we complement that study by investigating an orthogonal problem of developing better aggregation algorithms. For this, in our data-collection procedure, we obtain more annotations for each recording (seven vs. three) to give algorithm designers more freedom in using our data. Finally, to the best of our knowledge, the dataset collected by Novotney and Callison-Burch is not publicly available.

Another relevant contribution is a small dataset of translations from Japanese to English constructed by Li [28]. Each sentence in this dataset is associated with ten crowdsourced translations and a ground truth translation. We treat this data as a *baseline dataset* and return to it in Sections 5 and 6.

Several other works construct benchmark datasets for automated speech recognition without relying on crowdsourced annotation of audios. Specifically, LIBRISPEECH [37] (discussed in detail below) and GIGASPEECH [9] build on audios with known transcriptions (e.g., audio books or videos with human-generated captions). Starting from annotated long audios, they split the recordings into smaller segments and carefully align these segments with the ground truth texts. While this approach may result in high fidelity datasets, its applicability is limited to domains with pre-existing annotated recordings. Another clever approach is used in the COMMONVOICE dataset [2]. COMMONVOICE is constructed by starting from short ground truth texts and then crowdsourcing speech recordings of these texts. We note that this approach is complementary to ours (start from audios and then crowd-source transcriptions) and the choice between the two approaches may be application-dependent.

**Other Approaches** Several papers propose procedures to crowdsource high-quality annotations while avoiding automated aggregation of texts. One approach [26, 32] it to develop a multi-stage process in which initial transcriptions are improved in several rounds of post-processing. While this pipeline offers an appealing alternative to automated aggregation, it is much more complicated and may provide unstable quality due to variations in workers' accuracy. Another approach [5] is to reduce the aggregation of texts to the conventional classification problem. Specifically, given noisy annotations obtained in the first stage, a requester can hire an additional pool of workers to listen to original recordings and vote for the best annotation from the given set, thereby avoiding the challenging step of learning from noisy texts. However, this approach is associated with increased costs, and its accuracy is fundamentally limited by the accuracy of the best annotation produced in the first stage. In contrast, aggregation-based approach that we focus on in this work can in principle result in transcriptions that are better than all initial transcriptions of the corresponding recordings.

Table 1: Summary statistics for the source data used in this work. "Spkrs" stands for "speakers", letters M and F stand for male and female, respectively.

| Source Dataset | Version | Language | Nature | Total Length, hrs | # Recordings | # F Spkrs | # M Spkrs |
|---|---|---|---|---|---|---|---|
| LIBRISPEECH | train-clean | | | 38.8 | 11,000 | 52 | 46 |
| | dev-clean | | | 5.4 | 2,703 | 20 | 20 |
| | dev-other | English | Real | 5.3 | 2,864 | 16 | 17 |
| | test-clean | | | 5.4 | 2,620 | 20 | 20 |
| | test-other | | | 5.1 | 2,939 | 17 | 16 |
| RUSNEWS | Ru | Russian | Synthetic | 4.8 | 3,091 | 1 | 1 |

## 3  Data Source

In this section, we begin the description of our data collection procedure by introducing the pool of speech recordings that we annotate in Section 4. Table 1 gives an overview of our data sources.

**LibriSpeech Benchmark (cf. Contribution 1)** LIBRISPEECH is a famous benchmark for comparing speech recognition systems that consists of approximately 1,000 hours of read English speech derived from audiobooks and split into small segments (`https://www.openslr.org/12`). Specifically, LIBRISPEECH consists of two subsets — "*clean*" and "*other*". The *clean* subset contains recordings of higher quality with accents of the speaker being closer to the US English, while the *other* subset contains recordings that are more challenging for recognition. To achieve a better diversity of our data, we use both the *other* and *clean* parts of LIBRISPEECH: our initial pool of recordings comprises full dev and test sets as well as a part of the *clean* train set (11,000 recordings selected uniformaly at random from the *train-clean-100* subset of LIBRISPEECH). An important feature of the LIBRISPEECH dataset is its gender balance – approximately half of the recordings are made by female speakers. Thus, the recordings we use in this work are also gender-balanced.

**Under-Resourced Domains (cf. Contribution 2)** LIBRISPEECH is a rich source of recordings, but it is focused on a specific domain of audiobooks and contains recordings of English speech only. Thus, the aggregation algorithms trained on the annotations of LIBRISPEECH we collect in Section 4 may not generalize to other domains and languages due to potential differences in workers' behaviour. To alleviate this issue, we propose the following pipeline to obtain domain-specific datasets for fine-tuning or evaluating aggregation methods:

1. Obtain texts from the target under-resourced domain.

2. Use speech synthesis tools to construct recordings of texts collected in the previous step.

3. Obtain annotations of these recordings using crowdsourcing.[2]

The dataset constructed in this procedure can be used to fine-tune (evaluate) aggregation methods on data from novel domains. In this work, we demonstrate this pipeline by collecting a dataset of crowdsourced annotations of Russian speech recordings. Let us now describe the first two steps of the pipeline and introduce the second pool of synthetic recordings that we will annotate in Section 4.

*Texts from a Target Domain* For the sake of the example, we use the domain of news as our target domain. For this, we take sentences in Russian from the test set of the machine translation shared task executed as a part of the Eights and Ninth Workshops on Statistical Machine Translation [6, 7]. To support reliable evaluation, we additionally filter these texts to align their formatting with that used in the LIBRISPEECH dataset. A detailed description of this stage is given in Appendix C.

*Recording* Synthetic recording is the crux of our approach as it allows us to obtain recordings with known ground truth transcriptions without involving costly human speakers. In this example, we rely on Yandex SpeechKit [3] — an industry-level tool for speech synthesis — to obtain recordings of the ground truth texts. Importantly, Yandex SpeechKit gives access to both "male" and "female" voices, as well as to different intonations (neutral and evil). Thus, in the recording stage, we choose the "gender" and intonation for each recording uniformly at random, ensuring the diversity of our synthetic dataset.

---

[2]Note that this data captures the behavior of real workers in the target domain modulo potential differences induced by the use of a synthetic speech generator.

[3]`https://cloud.yandex.com/en-ru/services/speechkit`

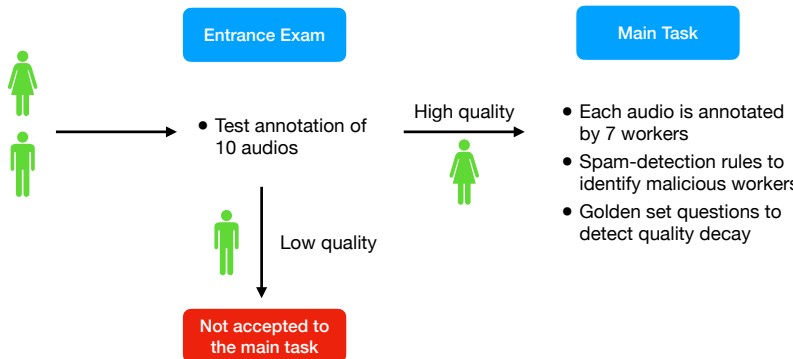

Figure 1: Schematic representation of the data annotation pipeline.

Following the procedure outlined above, we obtain 3,091 recordings of Russian speech that we title RUSNEWS. Table 1 gives summary statistics for two pools of recordings used in this work. In the next section, we will use audios from LIBRISPEECH and RUSNEWS to construct datasets CROWDSPEECH (based on LIBRISPEECH) and VOXDIY (based on RUSNEWS) of crowdsourced audio annotations.

## 4 Data Annotation

With datasets of speech recordings prepared in Section 3, we now proceed to the annotation stage in which we build our CROWDSPEECH and VOXDIY datasets. In that, we introduce the pipeline (Figure 1) that we used to gather reliable transcriptions on the Toloka crowdsourcing platform. [4] Additionally, throughout this section, we reflect on our data collection experience and give practical advice that may be useful for researchers and practitioners.

### 4.1 Task Design

We begin the exposition of our pipeline with a discussion of instructions, interface, and compensations.

**Instructions and Interface** Despite the task of audio transcription may sound natural to workers, there are important nuances that should be captured in the instructions and the interface. First, as our ground truth annotations have some specific formatting, we put a strong emphasis on conveying the transcription rules to the workers in the instructions. Next, throughout the task, workers may experience technical difficulties with some recordings, and we design the interface with that in mind. Specifically, at the beginning of the task, workers are asked whether the given audio plays well in their browser. The positive answer to this question triggers the text field, and the negative answer allows workers to report the technical issue without contaminating our data with an arbitrary transcription. The full version of instructions and a screenshot of the interface are given in Appendices A and B.

**Compensation** The recordings we annotate in this work can roughly be grouped by the level of difficulty: RUSNEWS and the *clean* subset of LIBRISPEECH are relatively easy while the *other* subset of LIBRISPEECH is harder to annotate. Thus, we issue a compensation of one cent (respectively, three cents) per annotation for recordings in the first (respectively, second) group. This amount of compensation was selected for the following reasons: (i) a typical amount of compensation for similar tasks on Toloka is one cent per recording; (ii) several past speech recognition studies that employed crowdsourcing [3, 31, 36] were issuing a compensation ranging from 0.5 cents to 5 cents for annotation of a recording of comparable length; (iii) a large fraction of workers on Toloka and other crowdsourcing platforms are residents of countries with a low minimum hourly wage.

**Workers Well-Being** Throughout the experiment, we were monitoring various quantities related to workers well-being. Specifically, the hourly compensation for active workers was close to or even exceeded the minimum hourly wage in Russia – the country of residence for primary investigators of this study. Additionally, our projects received mean quality ratings of 4.5 and above (out of 5) in anonymous surveys of workers, suggesting that workers deemed the work conditions reasonable.

---

[4]Code to fully reproduce the pipeline is available at `https://github.com/Toloka/CrowdSpeech`.

*Practical Comment.* In preliminary trials, we experimented with issuing compensations to workers even when they were unable to play the audio due to self-reported technical difficulties. Unfortunately, this setup resulted in workers reporting technical difficulties for a huge share of the tasks. Once we switched to the compensation for annotated recordings only, the amount of self-reported technical problems reduced drastically without affecting the quality of annotations. This observation suggests a spamming behavior in the original setup.

## 4.2 Worker Selection and Quality Control

Another key aspect of crowdsourcing data collection is to recruit the right population of workers for the task. For this, we make our task available to only those workers who self-report the knowledge of the language of the task: English for CROWDSPEECH and Russian for VOXDIY. Additionally, we implement an *Entrance Exam*. For this, we ask all incoming eligible workers to annotate ten audio recordings. We then compute our target metric — *Word Error Rate (WER)* — on these recordings and accept to the main task all workers who achieve WER of 40% or less (the smaller the value of the metric, the higher the quality of annotation). In total, the acceptance rate of our exam was 64%.

Importantly, to achieve a consistent level of annotation quality, it is crucial to control the ability of workers not only at the beginning but also throughout the task. For this, we implement the following rules to detect spammers and workers who consistently provide erroneous annotations:

- *Spam-detection rules.* Spammers often try to complete as many tasks as possible before getting detected and removed from the platform. To mitigate this behavior, we use a rule that automatically blocks workers from our projects if they complete two or more tasks in less than ten seconds.

- *Golden set questions.* We use golden set questions to continuously monitor the quality of annotations supplied by workers. If the mean value of the WER metric over the last five completed golden set questions was reaching $35\%$, we were blocking the worker from taking more tasks.[5]

*Practical Comment.* Workers may be hesitant to participate in the tasks if there is a risk that all their work is rejected without compensation. To avoid the additional burden on workers, we follow best practices of (i) compensating the exam for all workers who attempted it, irrespective of whether they passed the bar for the main task or not; (ii) issuing compensations to the workers for the tasks they have completed before being flagged by our quality control rules (these tasks are also included in the final datasets).

## 4.3 Running the Task

Having the pipeline prepared, we annotate each of the six sets of recordings described in Table 1 to construct our CROWDSPEECH and VOXDIY datasets. In that, we annotate each set of recordings in a separate pool (five pools for CROWDSPEECH with a common entrance exam and one pool for VOXDIY with a separate exam), keeping the task setup identical modulo the use of instructions in Russian for the VOXDIY dataset. Each individual recording was annotated by seven workers. If a worker reported technical issues on any recording, that recording was reassigned to another worker.

As a result of this procedure, we obtained six pools of annotated recordings — first five of these pools comprise the CROWDSPEECH dataset and the last pool of Russian recordings comprises the VOXDIY dataset. We release annotated data as well as the Python code to replicate our pipeline in the GitHub repository referenced on the first page of this manuscript.

*Privacy Remark.* The Toloka crowdsourcing platform associates workers with unique identifiers and returns these identifiers to the requester. To further protect the data, we additionally encode each identifier with an integer that is eventually reported in our released datasets.

# 5 Exploratory Analysis of Collected Datasets

Having constructed the CROWDSPEECH and VOXDIY datasets, we now proceed to the analysis of collected data on various dimensions, specifically focusing on the reliability of annotators. Before we delve into details, let us make some important remarks.

---

[5]For the sake of quality control, in this work we treat all recordings as golden set questions. In practice, one could annotate a handful of examples manually and use them as golden set questions.

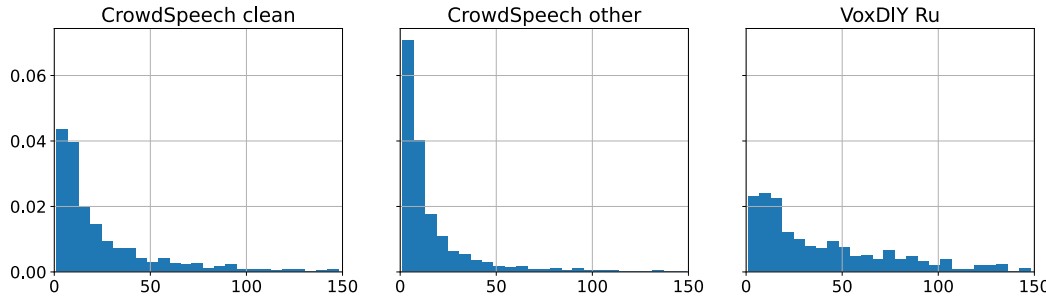

Figure 2: Distribution of the number of tasks completed by a worker. For brevity, we use only dev and test subsets of CROWDSPEECH and combine dev-clean and test-clean subsets (similarly, dev-other and test-other subsets) together.

First, for the sake of analysis, we slightly post-process the annotations obtained in our datasets by removing punctuation marks and making all sentences lowercased. This post-processing step is only needed to ensure consistency with the ground truth data but does not conceptually affect the quality of collected data. Second, when possible, we compare our datasets with CROWDWSA2019 — a dataset of crowdsourced translations (supplied with ground truth translations) constructed by Li et al. [23]. While this dataset is constructed in a different application, it is the largest publicly available dataset for the problem of noisy text aggregation. Hence, it is interesting to juxtapose it to our data. With these preliminaries, we are now ready to present the exploratory analysis of collected datasets.

## 5.1 Overview of Annotated Datasets

A general overview of the collected datasets is presented in Table 2. First, observe that in total, we have collected 176,519 annotations of 25,217 recordings made by 4,386 unique workers. Thus, our datasets are several orders of magnitude larger than CROWDWSA2019. To the best of our knowledge, our data is also the largest publicly available data of crowdsourced texts.

Second, it appears that in all datasets, the mean length of the crowdsourced annotations (translations) is slightly smaller than the mean length of the ground truth texts. This observation suggests that workers tend to skip some words in both the annotation and translation tasks.

Finally, Figure 2 shows the distribution of the number of tasks completed by a worker for data collected in this study. Observe that these distributions differ significantly between projects, likely being dependent on the task difficulty. It would be interesting to see if the aggregation algorithms can adapt for the changing distribution to provide a consistent improvement on different kinds of projects.

## 5.2 Inter-Rater Agreement

To evaluate whether the workers understood our task correctly, we compute Krippendorff's $\alpha$ [23], a chance-corrected inter-rater agreement measure that handles missing values and allows an arbitrary

Table 2: Overview of datasets collected in this work and comparison with CROWDWSA2019.

| Dataset | Version | Mean Sentence Length, words | | # Recordings | # Workers | # Answers |
| | | Ground Truth | Crowdsourced | | | |
| --- | --- | --- | --- | --- | --- | --- |
| CROWDSPEECH | train-clean | 34.6 | 32.8 | 11,000 | 2,166 | 77,000 |
| | dev-clean | 20.1 | 19.5 | 2,703 | 748 | 18,921 |
| | dev-other | 17.8 | 16.8 | 2,864 | 1,353 | 20,048 |
| | test-clean | 20.1 | 19.2 | 2,620 | 769 | 18,340 |
| | test-other | 17.8 | 16.8 | 2,939 | 1,441 | 20,573 |
| VOXDIY | RU | 13.8 | 13.6 | 3,091 | 457 | 21,637 |
| CROWDWSA2019 | J1 | 9.5 | 9.3 | 250 | 70 | 2,490 |
| | T1 | 11.9 | 9.1 | 100 | 42 | 1,000 |
| | T2 | 11.8 | 8.6 | 100 | 43 | 1,000 |

Table 3: Inter-rater agreement according to the Krippendorff's $\alpha$ with Levenshtein distance. Higher values indicate higher reliability.

| Dataset | Version | Overlap | Krippendorff's $\alpha$ |
|---|---|---|---|
| | train-clean | 7 | 0.81 |
| | dev-clean | 7 | 0.86 |
| CROWDSPEECH | dev-other | 7 | 0.77 |
| | test-clean | 7 | 0.84 |
| | test-other | 7 | 0.78 |
| VOXDIY | RU | 7 | 0.96 |
| | J1 | 10 | 0.37 |
| CROWDWSA2019 | T1 | 10 | 0.42 |
| | T2 | 10 | 0.41 |

distance function. In this work, we compute the value of Krippendorff's $\alpha$ using the Levenshtein distance (i.e., edit distance). Since the computation of $\alpha$ is time-consuming as it iterates over all the possible co-occurring item pairs, we obtain and report the sampling estimate of this value as follows. For each sample in the set of 10,000 samples, we randomly select 100 different audio recordings with replacement and compute $\alpha$ for all the transcriptions obtained for these recordings. We then take the mean of these values across all iterations and report it in Table 3.

Following recommendations of Krippendorff [23], we note that values of $\alpha \gtrsim 0.8$ suggest that annotations obtained for our dataset are reliable. Thus, we conclude that workers on Toloka successfully understood and performed our audio transcription task. Interestingly, the CROWDWSA2019 dataset demonstrates a much lower agreement between raters. We hypothesize that this discrepancy is due to the different natures of the tasks. Indeed, in the case of translations, there may be multiple equally good translations, and even ideal translators may have some disagreement. In contrast, the audio transcription task has unique underlying ground truth (ideal annotators can be in perfect agreement).

## 6 Evaluation

In this section, we evaluate the existing and novel methods for aggregation of noisy texts on our data. Specifically, in Sections 6.1 and 6.2 we analyze several existing methods. Next, in Section 6.3 we introduce a novel method developed on our data and compare it against the best baseline.

### 6.1 Baseline Methods

In our evaluations, we consider the following baseline methods, using implementations from the Crowd-Kit library (https://github.com/Toloka/crowd-kit) when available.

- **Random** A naive baseline that uniformly at random picks one of the annotations to be the answer.

- **ROVER** *Recognizer Output Voting Error Reduction* [18] was originally designed to combine the output of several different automatic speech recognition systems but was also demonstrated to work well on crowdsourced sequences [3, 17, 26, 31]. Under the hood, it aligns given sequences using dynamic programming and then computes the majority vote on each token.

- **RASA** *Reliability Aware Sequence Aggregation* [28] employs large-scale language models for aggregating texts. It encodes all worker responses using RoBERTa [30] (RuBERT[6] for the Russian language) and iteratively updates the mean weighted embedding of workers' answers together with estimates of workers' reliabilities. Finally, the method defines the final answer to be the response closest to the aggregated embedding based on the notion of cosine distance.

- **HRRASA** We also use a modification of RASA called HRRASA [27] that, besides the global reliabilities, uses local reliabilities represented by the distance from a particular response to other responses for the task. In the original paper [27], GLEU [33, 34] metric was suggested to calculate the distance between sequences, and we resort to this choice in our experiments as well.

---

[6]https://huggingface.co/DeepPavlov/rubert-base-cased

Table 4: Comparison of the baselines and the oracle performance. Evaluation criterion is the average word error rate (WER) and lower values are better.

| Dataset | Version | Oracle | Random | ROVER | RASA | HRRASA |
|---|---|---|---|---|---|---|
| CROWDSPEECH | dev-clean | 3.81 | 17.39 | 6.76 | 7.50 | 7.45 |
| | dev-other | 8.26 | 27.73 | 13.19 | 14.21 | 14.20 |
| | test-clean | 4.32 | 18.89 | 7.29 | 8.60 | 8.59 |
| | test-other | 8.50 | 27.28 | 13.41 | 15.67 | 15.66 |
| VOXDIY | RU | 0.70 | 7.09 | 1.92 | 2.22 | 2.20 |
| CROWDWSA2019 | J1 | 36.50 | 76.64 | 61.16 | 65.86 | 67.57 |
| | T1 | 28.07 | 63.08 | 51.35 | 48.29 | 49.99 |
| | T2 | 30.46 | 63.69 | 52.44 | 49.82 | 52.04 |

In addition to comparing the baselines, we want to make a rough conclusion on whether any of them demonstrate the optimal performance on our data. Note that it may be infeasible to uncover all transcriptions with absolute accuracy as for some recordings, the noise in annotations could vanish out all the signal. To obtain a more reasonable estimate of *achievable* performance, we introduce the **Oracle** aggregation algorithm to the comparison. For each recording, it enjoys the knowledge of the ground truth and selects *the best* transcription provided by the workers as its answer.

The Oracle method achieves the maximum accuracy that can be reached by an aggregation algorithm restricted to the set of transcriptions provided by the workers. Thus, Oracle gives a *weak* estimate of the *achievable* quality as its accuracy could be improved by an algorithm that is allowed to modify transcriptions provided by the workers. Nevertheless, in the analysis below, we focus on the *gap* between the baselines and the Oracle to estimate if there is some room for improvement on our data.

## 6.2 Performance of Baseline Methods

To evaluate baseline methods, we run them on each of the datasets under consideration excluding the train set of CROWDSPEECH as its main purpose is model training. We then compute the mean value of WER (Word Error Rate) over all recordings in each dataset and report it in Table 4. First, we note that when the quality of recordings is good, as in the case of the semi-synthetic VOXDIY dataset, non-trivial baseline methods achieve a near-perfect performance. This observation suggests that when the level of noise in collected annotations is relatively low, existing baselines satisfy the needs of practitioners.

However, observe that on the more challenging CROWDSPEECH dataset, there is a consistent gap between all baselines and Oracle, with the gap being larger for more difficult subsets (dev-other and test-other). This observation indicates that there is room for the development of better aggregation methods that keep up with, or even exceed, the performance of Oracle on more difficult tasks.

Finally, we note that the performance of all aggregation methods, including Oracle, is much weaker on the CROWDWSA2019 dataset. This effect is likely an artifact of the subjective nature of the machine translation task, which, in contrast to the speech recognition task, does not have a unique ground truth answer. Thus, CROWDWSA2019 may not be the best choice to design aggregation methods for objective tasks such as speech recognition. The same observation applies to the methods developed for that dataset (RASA and HRRASA): Table 4 indicates that on our data a simple ROVER baseline is always superior to these more advanced algorithms. Of course, a symmetric observation applies to our CROWDSPEECH and VOXDIY which may also be suboptimal for the machine translation task.

## 6.3 Novel Methods Developed on Our Data

In parallel with preparing this paper, we have designed and executed a shared task on developing aggregation methods for crowdsourced audio transcriptions [47]. The competitive nature of the task does not allow us to use LIBRISPEECH audios as it would result in the test data leakage. To avoid this problem, in the shared task we relied on the pipeline used to construct the VOXDIY dataset in the present paper. Specifically, we crowdsourced annotations of synthetic recordings of passages from Wikipedia and books [22].

Table 5: Comparison of the T5-based method developed on our shared task with ROVER — the strongest of the existing baselines. Models are compared on WER and lower values are better.

| Dataset | Version | Oracle | ROVER | T5 (ST) | T5 (ST+FT) | T5 (FT) |
|---------|---------|--------|-------|---------|-----------|---------|
| CROWDSPEECH | test-clean | 4.32 | 7.29 | 6.21 | 5.32 | 5.22 |
| | test-other | 8.50 | 13.41 | 11.80 | 10.46 | 11.67 |

One of the best results [39] in this shared task was demonstrated by a carefully fine-tuned T5 model [41]. With permission of the author, we evaluate their approach on the test sets of the CROWDSPEECH dataset collected in this paper. For this, we introduce three additional models (see details in Appendix D):

- First, we consider T5 model trained on the shared task data [**T5 (ST)**]
- Second, we fine-tune T5 (ST) on the development sets of CROWDSPEECH [**T5 (ST+FT)**]
- Third, we fine-tune the original T5 [41] on the train-clean subset of CROWDSPEECH [**T5 (FT)**]

Table 5 juxtaposes these models to the best of the available baselines (ROVER) on the test sets of CROWDSPEECH. First, we note that all T5-based models significantly outperform ROVER — the baseline that remained unchallenged for more than twenty years — setting new state-of-the-art results. Second, we observe that fine-tuning on the domain-specific data is crucial for our task as T5 (ST+FT) model is superior to T5 (ST) on both test sets. Similarly, T5 (FT) that got fine-tuned on the train-clean subset of CROWDSPEECH (which is much larger than dev-clean) outperforms other models on the test-clean set. Not surprisingly, however, it demonstrates lower accuracy on the test-other set because, in contrast to T5 (ST + FT), it did not get to see data from the *other* subset of CROWDSPEECH.

With these observation, we make the following conclusions:

- First, there is initial evidence that the data we release in this paper is instrumental in developing novel principled methods for aggregation of crowdsourced annotations.
- Second, observe that T5-based models do not use the fact that each worker provides multiple annotations in the dataset and do not estimate workers' expertise. In contrast, this information is known to be crucial for aggregation of categorical data [12, 50]. Thus, we believe that it is possible to further improve the results by coupling T5 with some way to estimate worker skills.
- Third, note that the T5-based method is designed on the data collected through the semi-synthetic procedure used to construct VOXDIY. Given that the strong performance of thes method carried over to the realistic CROWDSPEECH dataset, we conclude that the semi-synthetic data may be useful to quickly explore new domains in which no annotated recordings of human voice exist.

Finally, we refer the reader to Appendix E where we provide additional comparison of models in terms of types of errors they make.

# 7   Conclusion

In this work, we collected and released CROWDSPEECH — the first publicly available large-scale dataset of crowdsourced audio transcriptions. Based on evaluations of existing and novel methods on our data, we believe that our work will enable researchers to develop principled algorithms for learning from noisy texts in the crowdsourcing setting. Additionally, we proposed an automated pipeline for collecting semi-synthetic datasets of crowdsourced audio transcriptions in under-resourced domains. We demonstrated this pipeline by constructing VOXDIY — a Russian counterpart of CROWDSPEECH.

In the end, we should mention some limitations of our work. First, we admit that the use of speech synthesis techniques could affect the distribution of errors people make when annotating audios, thereby affecting the generalization ability of aggregation tools trained on VOXDIY. Second, in this work, we annotated our datasets in an industry-level pipeline, which resulted in annotations of high quality. It would be interesting to additionally collect datasets under less stringent quality control rules or for more challenging data to make our data even more diverse in terms of complexity.

Better understanding and addressing these limitations is an interesting direction for future research. With these caveats, we encourage researchers and practitioners to use our data judiciously and to carefully evaluate all the risks and benefits in their specific application.

## Acknowledgements & Author Contributions

I.S. and N.P. designed the setup of the problem and the data-collection pipeline. N.P. collected data on Toloka. N.P. and D.U. conducted data analysis. All authors contributed to the writeup.

The work of I.S. was supported in part by NSF grants CIF 1763734 and CAREER: CIF 1942124.

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
