# OpenReview forum: "CrowdSpeech and Vox DIY: Benchmark Dataset for Crowdsourced Audio Transcription"
_NeurIPS.cc/2021/Track/Datasets_and_Benchmarks/Round1 — NeurIPS 2021 Datasets and Benchmarks Track (Round 1)_

### Official Review · Reviewer_fKcU · 2021-06-29
**Benchmarking Crowdsourced Audio Transcriptions**

**Rating:** 7
**Confidence:** 4
**Correctness:** I did not find any methodological iss…

**Strengths:**

The datasets presented in the paper are filling some important gaps. With these datasets, one can check new, more advanced methods for transcription aggregation.

It's good that the authors compared their datasets to the CrowdCSA2019 data for Machine Translation. It was instructive to see the similarities and differences between them.

The authors present some interesting insights, e.g. the one about the length of crowdsourced annotations (280).


**Weaknesses:**

The main weakness, as the authors themselves notice, is that the VoxDIY is too easy, it should be definitely augmented with realistic noise. And in general, the fact that the data set was synthesized is somewhat problematic.

It would be interesting to see more baselines, even simple ones, e.g. the longest sentence, the most frequent sentence (and switch to random if no sentence is repeated; actually it would be interesting to see statistics for sentence repetitions) - though this is not really an important weakness.



**Additional Feedback:**

Minor remarks:

* the title should be broken before "for" (Dataset \\ for)
* 242: "integral number" => "integer", just "number"/"identifier"?
* the numbers in tables are not aligned
* are references [5] & [6] correct?

**Clarity:**

The paper is well written, the quality of presentation is high. I have no remarks as far this aspect is concerned.

**Documentation:**

The data set is well documented and organized. No remarks here.

**Ethics:**

I did not find anything raising any ethical concerns.

**Relation To Prior Work:**

I did not find any obvious missing references. Maybe, some tasks other than ASR and MT should be covered? See e.g. Clematide, Simon, Lenz Furrer, and Martin Volk. "Crowdsourcing an OCR gold standard for a German and French heritage corpus." (2016): 975-982.

**Summary And Contributions:**

This is a sold and well-written, though not ground-breaking, work on creating benchmarks for measuring the quality of crowdsourced audio transcriptions.

The contributions of this papers are as follows:

* CrowdSpeech - a large dataset for crowdsourced audio annotations (English)
* VoxDIY - a similar dataset for Russian (though based on synthesized, not real audio recordings)
* evaluation of several baseline methods for aggregation of crowdsourced transcriptions
* methodology and an open-sourced pipeline for preparing such datasets

---

### Official Review · Reviewer_xn39 · 2021-07-01
**A relevant new audio to text dataset.**

**Rating:** 7
**Confidence:** 3
**Correctness:** The work seems correct and constructe…
**Clarity:** Yes.

**Strengths:**

- The provided datasets enrich the field of audio to text transcriptions.
- Both code and data are available and there seem to be no ethical questions.
- A method on how to extend audio data for underrepresented domains is discussed.



**Weaknesses:**

- For Russian, texts are used and automatically converted to audio. This method is limited and not representative of human diversity when reading or speaking. For this reason, those data are not comparable to the remaining datasets.
- The work does not provide results on benchmarking Machine Learning tasks.


**Additional Feedback:**

No additional feedback.

**Documentation:**

Yes.

**Ethics:**

No.

**Relation To Prior Work:**

Yes.

**Summary And Contributions:**

The paper presents the annotation of two existing datasets in terms of audio transcription by 7 workers. The first dataset for the English language corresponds to more than 20 hours of audiobooks (LibriSpeech) that crowd workers transcribe to text. The second dataset for the Russian language starts with existing news texts. Those are automatically transcribed to audio and then crowd workers transcribe it again to text.

The related work section supports the need for new larger datasets and the discussion on the different aggregation methods for different crowdsourced replies. Also, previous datasets are described in this section. One of those (CROWDCSA2019) is used later for comparison with the new annotated datasets.

The annotation is described in detail and exploratory analysis of the collected datasets proves the quality of the annotated data. For instance, Krippendorff's alpha between annotators proved agreement with a value around 0.8.

For evaluation, the paper compares the existing methods for aggregation of noisy texts on the new datasets. ROVER, RASA, HRRASA, and Oracle are computed. When compared with the previous datasets the new annotated material seems to provide better quality for all metrics.

---

### Official Review · Reviewer_tHBQ · 2021-07-04
**A dataset of crowdsourced audio transcriptions for evaluating aggregation methods.**

**Rating:** 6
**Confidence:** 4

**Strengths:**

- This paper releases a dataset of speech transcriptions that can foster research on aggregation methods for audio transcriptions.
- It provides a common benchmark for evaluating such algorithms.
- The paper provides useful guidelines on how the current aggregation methods work and, more in general, on how a high-quality crowdsourced data collection should be performed.

**Weaknesses:**

- The paper releases a dataset potentially valuable for part of the community but it focuses on a "niche" problem (i.e, aggregation of speech transcriptions from crowdsourced workers ) that might be of limited research interest. As a matter of fact, the best aggregation is still Rover (an old method based on dynamic programming).

- The author mentioned that crowdsourced data might be useful to fine-tune a speech recognizer to adapt it to new conditions like new accents, etc.  In the paper, I would thus have expected some real speech recognition experiments where the different aggregation algorithms are compared using the final speech recognition performance achieved after fine-tuning the model.

- The author claims: "The main obstacle towards designing advanced aggregation methods is the absence of training data, and in this work, we focus on bridging this gap in speech recognition".  If I'm not misinterpreting this sentence, it looks like the authors suggest that CROWDSPEECH can be used to train an aggregation method. I would thus have expected some initial attempts in this direction (e.g, a seq2seq neural network that takes in input the multiple transcriptions and outputs the combined one).

- CROWDSPEECH is based on LibriSpeech which is a very specific dataset based on old audiobooks. It looks like this task is very challenging for workers, as witnessed by the high WER results achieved with all the considered aggregation methods. I'm not really sure if the results achieved with CROWDSPEECH actually hold in a more realistic crowdsourcing scenario.

- As mentioned by the authors, I'm very skeptical as well on the results achieved with synthetic speech.  IMO these results don't add too much to the paper and, instead, raise some doubts on their real significance. I would even consider removing them.

- It could be great to add an error analysis to highlights where most of the errors done by the users and by the aggregation methods come from.

- "Vox Populi, Vox DIY: Benchmark Dataset for Crowdsourced Audio Transcription": I don't understand why the title mentions "Vox Populi". This name is not reported in any other place of the article. Moreover, voxpopuli is the name of another popular dataset recently released (https://arxiv.org/abs/2101.00390). Please, explain this.

**Additional Feedback:**

I shared all my feedback and comments above. In summary, I think the paper provides some contribution valuable to a part of the community, but it is still doesn't match the quality needed in a top conference like NeurIPS.

**Clarity:**

The paper is not always clear. I had to read it carefully 2-3 times to understand it more in detail. For instance, the connection between CROWDSPEECH => Librispeech and RUSNEWS=>VOXDIY should be made clearer from the beginning.

**Correctness:**

The dataset is constructed soundly and the methodology used to collect data is explained.  The experiments on the proposed datasets are conducted on some popular aggregation methods. As mentioned above, I would have expected experiments on speech recognition and experiments on neural aggregation methods.

**Documentation:**

The transcriptions provided by the users are available here (along with the tools needed to replicate the experiments):
https://github.com/pilot7747/VoxDIY/
The documentation + code comments can be improved but at least I'm able to find easily the most important contribution (i.e., the transcriptions on librispeech)

**Ethics:**

I don't see ethical issues with this specific paper.  A general concern with crowdsourcing platforms is that workers are often underpaid.

**Relation To Prior Work:**

I think references should be improved for this paper:
- The author should cite and discuss CommonVoice (https://commonvoice.mozilla.org/) that it is also a very popular system for collecting transcribed audio.
- The author should cite and discuss GigaSpeech (https://arxiv.org/pdf/2106.06909.pdf) where the authors propose a way to collect annotated data from the web.
- "The research community has been actively developing tools for automated speech recognition [1, 18, 24 27, 30, 35]" => This looks like a list of random speech papers. I would suggest being much more complete  here and cite, for instance, the numerous open-source speech toolkits largely used in the community (not just DeepSpeech)

**Summary And Contributions:**

The paper focuses on the problem of aggregating transcriptions of speech signals from crowdsourced workers. In practice, multiple workers are asked to transcribe the same speech sentence.   An aggregation algorithm should try to retrieve a more reliable transcription (hopefully closer to the real ground truth) by combining users' transcriptions. The authors collected a dataset of transcriptions called CROWDSPEECH that is based on the manual annotation of the dev and test subsets of LibriSpeech. This dataset can be used as a benchmark to evaluate aggregation algorithms. The authors also created a dataset based on a synthesized speech called RUSNEWS  which is released with the corresponding annotations from the workers (under the name Vox DIY).
Finally, the authors evaluate some aggregation methods (i.e. ROVER, RASA HRRASA) on the proposed dataset.

---

### Decision · Program_Chairs · 2021-07-26

**Decision:**

Accept

**Comment:**

This paper makes two contributions:
(1) collecting crowdsourced transcripts for existing speech data (LibriSpeech) and synthesized speech data (RusNews). The novel aspect is that each audio is annotated by 7 different crowd workers, providing the largest publicly available dataset for crowdsourced texts aggregation (99K annotations of 11K recording by 3K unique workers).
(2) evaluating different methods (RASA, HRRASA, T5) for aggregating sequence level annotation from crowd workers.

The data collection method follows standard practices and shows high inter-annotator agreement (table 3) and will be useful for the research community studying imperfect annotations. The comparison between transcript for real speech (LibriSpeech) and synthesized speech data (RusNews) is also interesting. Overall, the paper is well written, and the newly added T5 baseline results (outperforming other aggregation methods) are interesting.